# Monte Carlo dosimetric characterization of the IRAsource high dose rate Iridium-192 brachytherapy source: Comparison with the mHDR-v2r model

Shuhei Tsuji[iD]*

Natural Sciences, Kawasaki Medical School, Matsushima, Kurashiki, Japan

* tsuji@med.kawasaki-m.ac.jp

## Abstract

The aim of this study was to evaluate the dosimetric characteristics of an IRAsource brachytherapy source using Monte Carlo simulations incorporating source electron effects. The dose rate constant, radial dose function, and two-dimensional (2D) anisotropy function were calculated and compared with those of published data and results of the mHDR-v2r source model. The dose rate constant obtained for the IRAsource was $(1.1142 \pm 0.0030)$ cGyh$^{-1}$U$^{-1}$, which was consistent with previously reported values, within the range of experimental uncertainty. The radial dose function exhibited a pattern similar to that of the mHDR-v2r source, with dose distributions converging at distances of approximately 10 cm, where the effect of capsule thickness became negligible. The 2D anisotropy function did not fully align with other experimental datasets for the IRAsource. Even when comparing the IRAsource with mHDR-v2r sources, the 2D anithotropy function showed a difference of 3% to 6% around 0 degrees at distances of 0.25 cm to 5.0 cm from the source. The results reflected reasonable trends that were consistent with differences in source capsule geometry of the IRAsource and the mHDR-v2r. These findings show that the dosimetric data presented here is reliable. This study provides essential baseline data for accurate dose assessment of the IRAsource in brachytherapy and underscores the importance of further experimental validation to refine dose characterization.

## Introduction

Brachytherapy is a crucial form of radiation therapy for cancer treatment that involves the direct delivery of high doses of radiation to localized tumors. This method minimizes the impact on surrounding healthy tissue while achieving effective treatment through the placement of a radiation source close to the tumor. The TG-43U1 protocol, published by the American Association of Physicists in Medicine (AAPM), includes guidelines concerning the standardization of radiation calculations in

**Data availability statement:** All relevant data are within the manuscript and its Supporting information files.

**Funding:** The author(s) received no specific funding for this work.

**Competing interests:** No competing interests exist.

brachytherapy [1]. These guidelines provide essential information regarding the characteristics of radiation sources and the evaluation of dose distributions, particularly in brachytherapy. Various sources are used for treatment planning according to the TG-43U1 protocol. For each source, the dose rate constant, radial dose function, and two-dimensional (2D) anisotropy function are determined using Monte Carlo (MC) simulations [1–13].

In 2015, the Nuclear Science and Technology Research Institute in Iran developed a new high dose rate (HDR) $^{192}$Ir brachytherapy source model called IRAsource [14]. The aim of this innovative method was to enhance the precision and effectiveness of brachytherapy for cancer treatment.

The development of an IRAsource can improve the quality of cancer treatment and contribute to international brachytherapy research. The source dimensions are shown in Fig 1(a), revealing that the dimensions of $^{192}$Ir in IRAsource are identical to those of mHDR-v2r [2]. Therefore, its validity can be evaluated by comparing it with findings from previous studies on mHDR-v2r [3]. The thickness of the capsule at the top of the source for the IRAsource is 0.53 mm, which is more than twice that of the mHDR-v2r (0.2 mm). Therefore, the absorbed dose of the IRAsource is expected to differ from that of the mHDR-v2r at angles near $\theta = 0°$ but shows a similar trend at angles near $\theta = 90°$. The parameters for the IRAsource TG-43U1 protocol have been previously described [12,13]. However, compared with the parameters of mHDR-v2r [2,3], these parameters are not similar and deviate completely. The aim of this study was to evaluate the dosimetric characteristics of an IRAsource brachytherapy source using MC simulations, incorporating source electron effects. The Electron-Gamma Shower Version 5 (EGS5) code was used for the MC simulations. MC simulations were performed not only with photons but also with electrons. Based on the results of this MC simulation, the TG-43U1 protocol parameters of the IRAsource were determined and compared with those of mHDR-v2r.

## Materials and methods

### EGS5 and EGS5MPI code

In the EGS5 code system [15], which was used as the MC simulation code, a double random hinge approach is used to effectively separate energy loss and multiple elastic scattering to model the spatial transport of electrons and positrons. Within the EGS5 MC framework, regions (referred to as tallies in other MC codes) are defined through combinatorial geometry (CG) routines. The material information was generated using the Preprocessor for EGS (PEGS), a set of subprograms embedded in EGS5. To execute the simulation, specifying the geometric configuration of regions (tallies) and the PEGS-generated material data associated with them is necessary. The SHOWER subroutine is used to define the initial particle parameters, such as charge (0: photon, −1: electron, +1: positron), energy, position, direction, region, and weight. Particles are tracked until they leave all the region or their energy decreases below the cutoff value. The AUSGAB subroutine is triggered under specific conditions, including transport over a given distance (0), sub-threshold energy (1, 2), user-requested termination (3), or partial energy deposition from binding effects (4). This

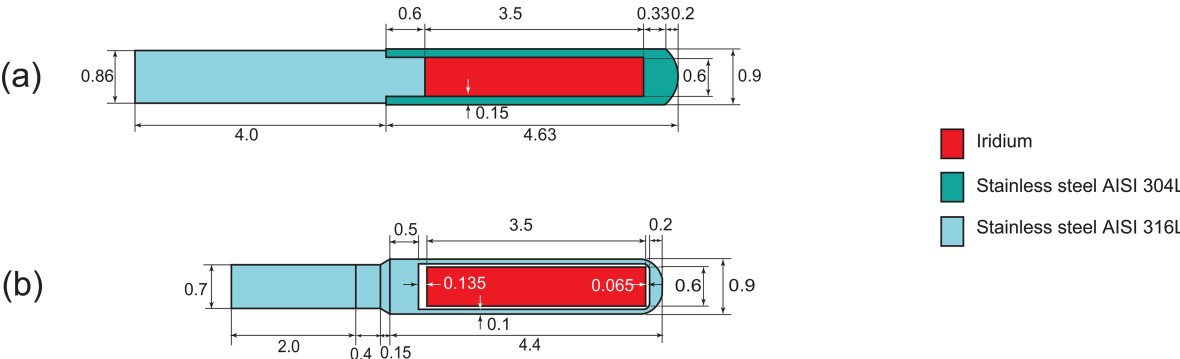

**Fig 1**. **Schematic designs of (a) IRAsource and (b) mHDR-v2r.** Dimensions are given in milimeters. The IRAsource cable and capsule density was 8.02 gcm$^{-3}$, while the mHDR-v2r capsule density was 8.02 gcm$^{-3}$ and the cable density was 4.81 gcm$^{-3}$.

routine collects and processes the relevant physical quantities carried by particles. Step size control is provided via the "characteristic dimension" (CHARD), which is typically chosen as the smallest characteristic length of the region. This calculation was performed with the EGS5 code in parallel using the message-passing interface (EGS5MPI) technique [16]. Photons and electrons were simulated independently as particles from a radiation source. The following parameters were used in various regions or media: sampling of the angular distributions of photoelectrons, K- and L-edge fluorescent photons, K and L Auger electrons, Rayleigh scattering, linearly polarized photon scattering, incoherent scattering, and Doppler broadening of the Compton scattering energies.

## Description of source materials

The shape of the IRAsource is illustrated in Fig 1(a). The dimensions and materials of the IRAsource used in the simulation were obtained from studies by Ayoobian et al. [14] and Sarabiasl et al. [12]. The angle from the source center was consistent with the AAPM TG-43U1 protocol. The composition weight ratios of the capsule were 19.04% Cr, 71.82% Fe, 8.32% Ni, 0.22% Cu, and 0.62% Mo, and those of the cable were 17.21% Cr, 69.77% Fe, 10.17% Ni, 0.36% Cu, and 2.52% Mo. The source cable length was 4 mm. The densities of the capsules and cables were set to 8.02 gcm$^{-3}$. On the other hand, the mHDR-v2r capsule and cable were made of the same stainless steel AISI 316L as the IRAsource cable, but the composition was slightly different (the composition weight ratios: 2% Mn, 1% Si, 17% Cr, 12% Ni, and 68% Fe) [2]. The densities of the capsule and cable were 8.02 gcm$^{-3}$ and 4.81 gcm$^{-3}$, respectively, and the length of the cable was 2 mm.

## Photon and electron spectra

The $^{192}$Ir photon spectrum produced by the National Nuclear Data Center [17,18], which was subsequently quoted by Rivard et al. [9], was inputted into the MC simulations (total photon spectra = 2.2992 photons/Bq, with a cutoff energy of 10 keV). The electron spectrum included $\beta$ decay. As recommended by the International Commission on Radiological Protection [19,20], internal conversion electrons were used in the MC simulation. The sum of the continuous spectra of $\beta$ decay with a cutoff energy of 10 keV was 0.9192 electrons/Bq (energy bin width $\times$ differential spectrum), whereas that of the internal conversion electrons with a cutoff energy of 10 keV was 0.1531 electrons/Bq.

## MC methods of dose rate

Photons and electrons were independently simulated as particles from a radiation source with a cutoff energy of 10 keV. The absorbed dose rate was used to summarize the energy deposit of each particle emitted from a source without energy

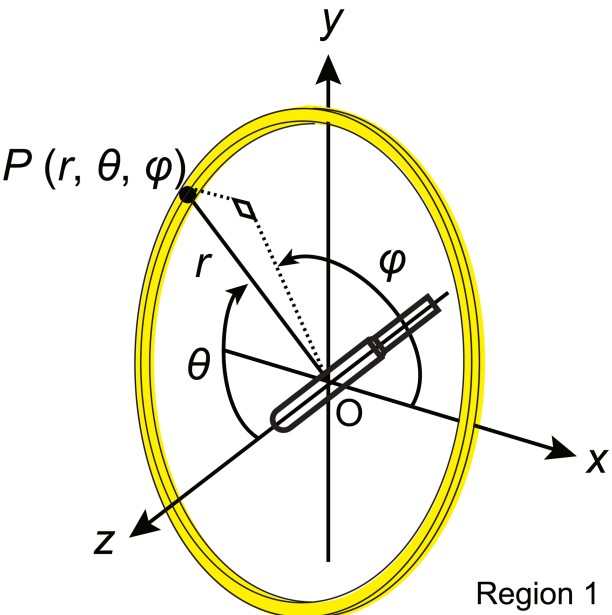

**Fig 2. MC coordinate system.** $r$, $\theta$ and $\phi$ of $P$ are assumed, as shown in the figure. The data acquisition area is shown in yellow. This figure refers to Fig 4 in Reference [3].

fluences or mass-energy absorption coefficients. The diameter of the water sphere phantom was set to 40 cm. In the phantom, pure water without gas was used, as recommended by TG-43U1 [1], at a density of 0.998 gcm$^{-3}$.

As illustrated in Fig 2, the coordinates indicate the longitudinal axis of the radiation source along the z-axis, with the direction of the source tip extending from the cable taken as positive. The origin of the coordinate system was at the center of the source. In the polar coordinate system, $r$, $\theta$, and $\phi$ denote the radius, polar angle, and azimuthal angle of point of interest $P$, respectively. The volume $V(P)$ of the sliced area at $P(r, \theta, \phi)$ was integrated from $r - \Delta r$ to $r + \Delta r$ and from $\theta - \Delta \theta$ to $\theta + \Delta \theta$. This sliced region, highlighted in yellow in Fig 2, represents the target volume of $P$. Energy deposition within the yellow-highlighted region was aggregated. The absorbed dose rate $\dot{D}$ at point $P$ was obtained by dividing this sum by the mass of the region, the number of source photons and electrons, and the emission rate of the radiation source. Values for $\dot{D}$ at $P$ were tabulated from 0.05 to 20 cm. The aggregation width, $\Delta r$, was 0.0025 cm from 0.05 to 1.5 cm and 0.025 cm from 2 to 20 cm. The angles $\dot{D}$ were set from 0° to 180° in increments of 1°. For $0° < \theta < 180°$, the aggregation width was $\Delta \theta = 0.1°$ (0.001745 rad), whereas that for $\theta = 0°$ or 180°, it was $\Delta \theta = 0.5°$ (0.08727 rad). Data were aggregated using $\theta$ to $\theta + \Delta \theta$ or $\theta - \Delta \theta$.

Since the minimum bin width in water is $2 \times \Delta r = 0.005$ cm, CHARD in the water region of the EGS5 program was set to 0.005.

Fig 3 shows the step length distribution of electrons from the MC simulation when electrons with a total energy of 1.180 MeV and photons with an energy of 1.378 MeV are incident on water when CHARD in the program is set to 0.005. These energies are the maximum values of the energy spectrum used in this MC simulation. Each distribution has peaks in bins from 0 to $1.0 \times 10^{-4}$ cm. Therefore, the bin width $\Delta r = 0.0025$ cm for determining the energy deposit is sufficiently large.

Overall, $1.0 \times 10^{11}$ events were calculated for photons and electrons for each source, such that the standard error of the absorbed dose at $r = 1$ cm and $\theta = 90°$ was 0.2%.

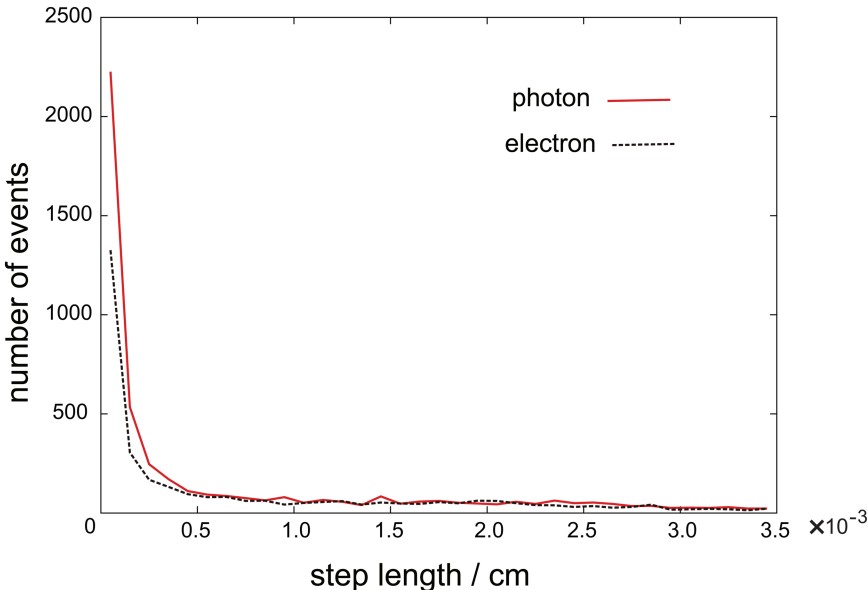

**Fig 3. Distribution of electron step lengths obtained from the MC simulation when electrons and photons with energies of 1.180 MeV and 1.378 MeV are projected into water.** The number of events projected into water was 15 for each. The total number of step lengths was 3402 for electrons and 4932 for photons. The characteristic dimension (CHARD) was set to 0.005.

## MC methods of air kerma strength

To determine the air kerma, $K(r)$, at the transverse-axis distances $r$, the kinetic energy transferred from photons to electrons within a small region of air was summed using MC simulations and divided by the mass of that volume. The air kerma rate, $\dot{K}(r)$, was calculated from $K(r)$, considering the number of photons and emission rate of the radiation source. The air kerma strength, $S_K$, was obtained by fitting a linear function to the values of $\dot{K}(r) \cdot r^2$ for each $r$. Spherical shell phantoms with various radii were used to compute $K$, with radii spaced at 10 cm intervals from 10 cm to 120 cm from the center of the source. Each air layer had a thickness of 1 mm, and the surrounding space was maintained under vacuum. For each value of $r$, the data aggregation window was set at $\Delta r = 0.05$ cm and $\Delta\theta = 1°$ (0.01745 rad) at $\theta = 90°$. Air was modeled as dry (0% humidity) following the recommendations of AAPM Report 229 [10], with elemental weight fractions of nitrogen (N), 75.5%; oxygen (O), 23.2%; and argon (Ar), 1.3%. The air density was 0.00120 g/cm³. Only photons with energies above the 10 keV cutoff were generated from the source. CHARD in the air region of the EGS5 program was set to 0.1. In total, $1.0 \times 10^{12}$ events were simulated to ensure a standard error of $\leq 0.3\%$ at $r = 10$ cm. A more detailed analysis method can be found in Reference [3].

## Results

The mHDR-v2r dosimetric values were taken from Reference [3] and compared with those of the IRAsource dosimetry.

## Contribution of source electron and gamma of each source

The contributions of photons and electrons in the immediate environment of the IRAsource were investigated (Figs 4 and 5). For comparative analysis, the results obtained using mHDR-v2r were also examined [3].

Between 0.5 and 1.5 mm, the values for photons and electrons were lower from the IRAsource than from mHDR-v2r. One possible explanation is that, although the source sizes are identical, the capsule thickness at $\theta = 90°$ was 0.05 mm greater for the IRAsource.

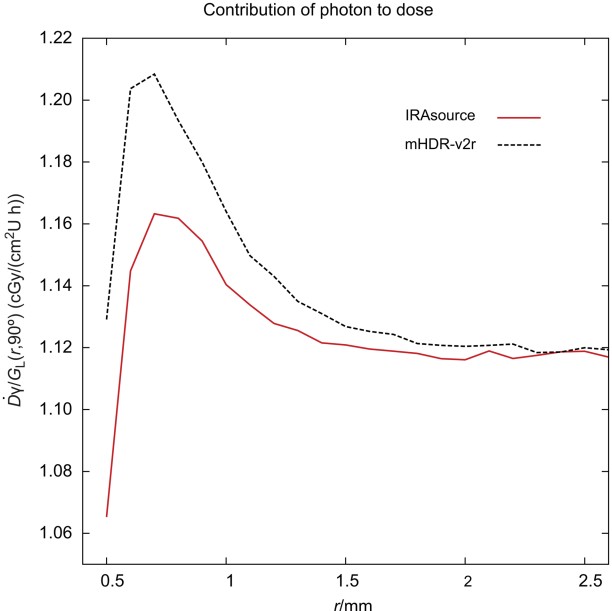

**Fig 4. Absorbed dose rate from source photons near the source.** The absorbed dose rate was normalized by the unit of air kerma strength and the geometry function.

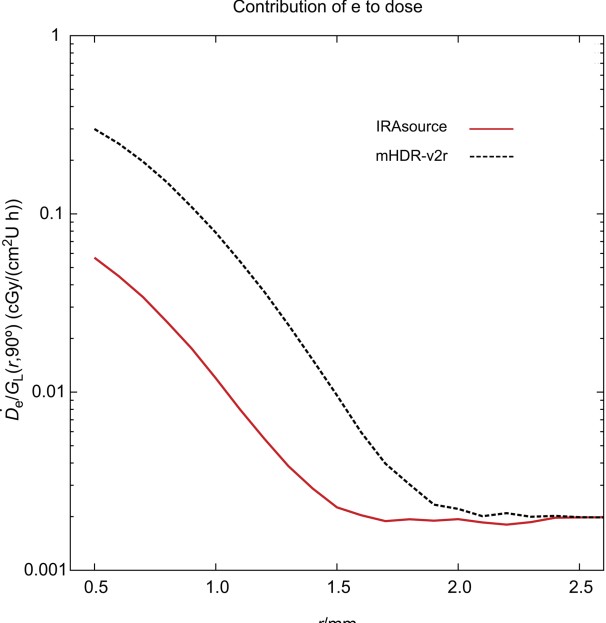

**Fig 5. Absorbed dose rate from source electrons near the source.** The absorbed dose rate was normalized by the unit of air kerma strength and the geometry function.

### TG-43U1 dataset: Dose rate constant

The dose rate constant for the IRAsource was determined using the air-kerma strength and the absorbed dose at $r = 1$ cm, $\theta = 90°$. The resulting dose-rate constant for the IRAsource was as follows:

$$\Lambda = (1.1142 \pm 0.0030) \text{ cGyh}^{-1}\text{U}^{-1}$$

According to previous studies, the dose rate constant reported by Sarabiasl et al. is $(1.112 \pm 0.005)$ cGyh$^{-1}$U$^{-1}$, whereas Rostami et al. reported a value of 1.1094 cGyh$^{-1}$U$^{-1}$. Ayoobian et al. reported a corresponding calculated value: $(1.112 \pm 0.008)$ cGyh$^{-1}$U$^{-1}$ using MCNP4C. The value obtained from the MC simulation was higher than that of the other MC simulations. Ayoobian et al. also reported two experimental results: $(1.129 \pm 0.044)$ cGyh$^{-1}$U$^{-1}$ using HD-810 film, and $(1.084 \pm 0.046)$ cGyh$^{-1}$U$^{-1}$ using EBT film. The results of this MC simulation were within the standard error margin of reported values.

### TG-43U1 dataset: Radial dose function

The radial dose function $g_L(r)$ of the IRAsource is presented in Fig 6. This figure also includes the results reported by Ayoobian et al. [14], Sarabiasl et al. [12], and Rostami et al. [13], as well as those for the mHDR-v2r source [3]. The steep increase near the IRAsource was not as pronounced as that observed for the mHDR-v2r.

Fig 7 shows the ratios of the $g_L(r)$ from other studies compared with those obtained in this study. The MC simulation results from other studies for the IRAsource [12–14] were lower than those obtained in this study, whereas the experimental measurements [14] were higher than all the simulated results. Notably, the results of this study were intermediate between those of other simulations and the experimental data. The findings were most different from those of Rostami et al., but the difference was within 2% from 1 cm to 10 cm. At approximately 10 cm, the results closely matched those of the mHDR-v2r [3].

### TG-43U1 dataset: 2D anisotropy function

Figs 8 through 11 present the results of the 2D anisotropy function. Each figure includes the results of this study and those reported by Rostami et al. [13] and Sarabiasl et al. [12]. for IRAsource, as well as the results for the mHDR-v2r source [3]. Furthermore, the figures display the ratios of the mHDR-v2r [3] results to those of the IRAsource. The angles reported by Sarabiasl et al. were converted using the TG-43U1 protocol [1].

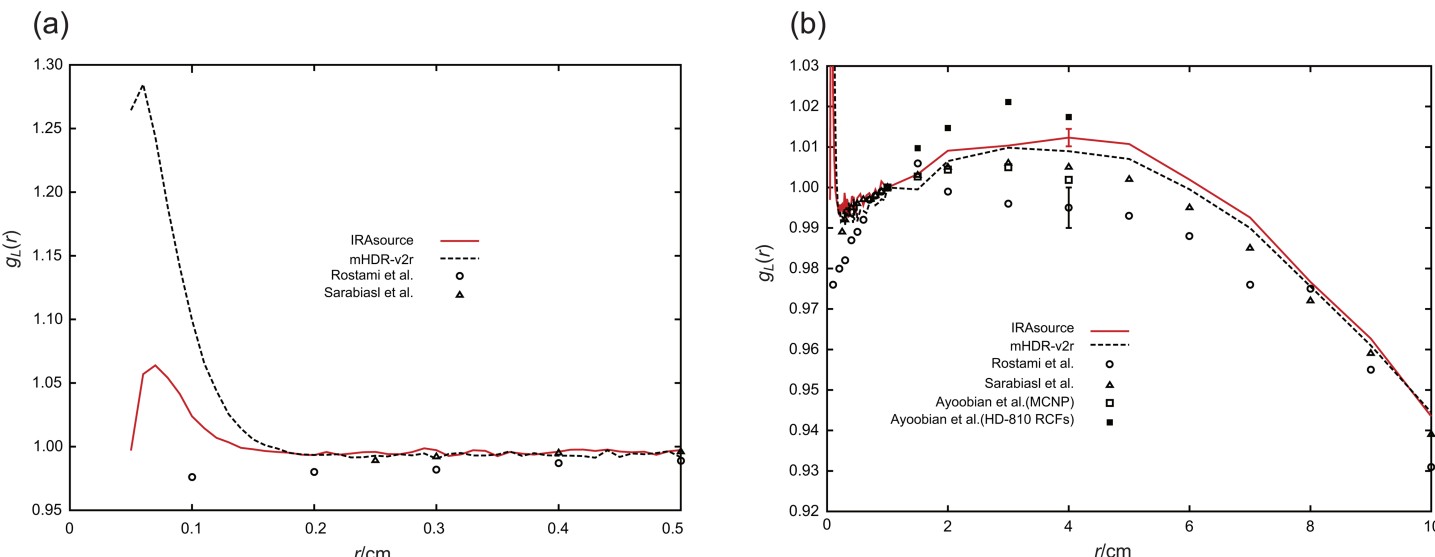

**Fig 6. Comparison of the radial dose functions.** (a) 0 cm $\leq r \leq$ 0.5 cm. (b) 0 cm $\leq r \leq$ 10 cm. The standard error was used at $r = 4$ cm.

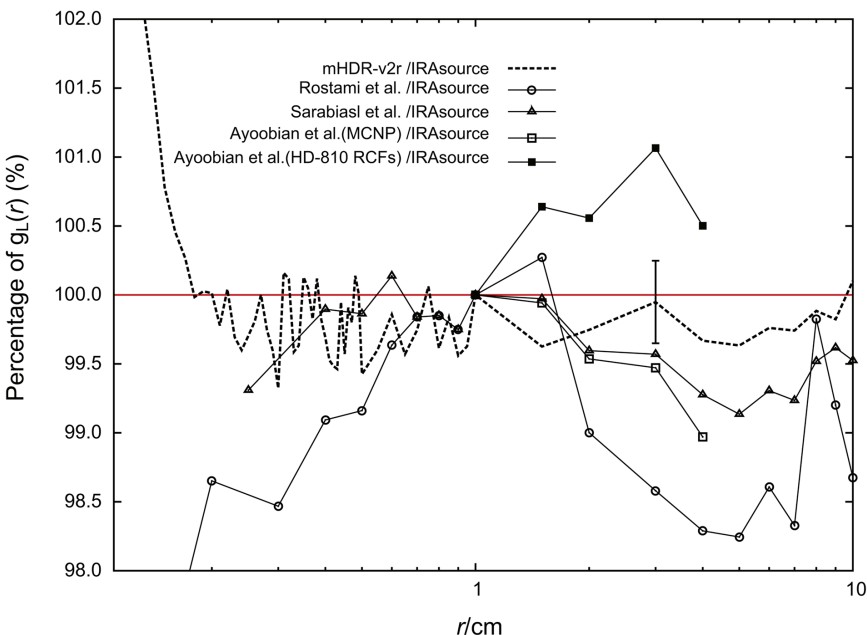

**Fig 7**. **The ratio of the radial dose functions to the IRAsource.**

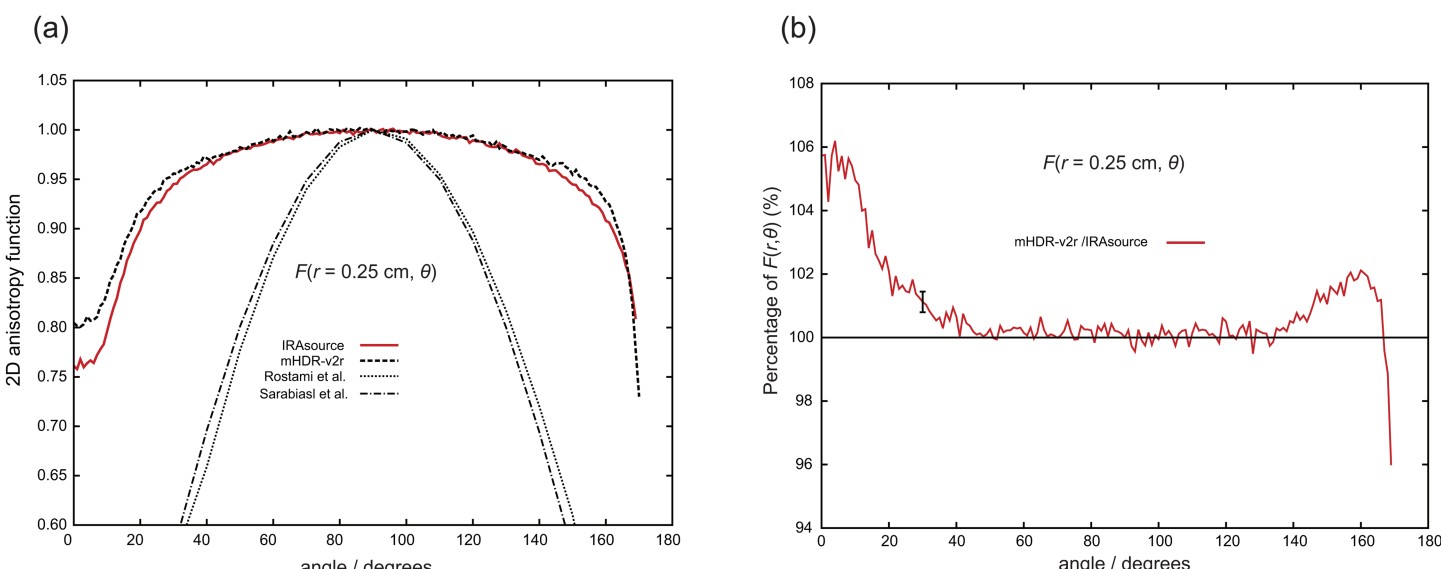

**Fig 8**. **(a) Comparison of two-dimensional (2D) anisotropy functions at distances of 0.25 cm.** (b) The ratio of 2D anisotropy functions to IRAsource at 0.25 cm. The error bar at 30° was derived from the relationship between IRAsource and mHDR-v2r [3].

A comparison with mHDR-v2r reveals that across these figures, mHDR-v2r may exhibit higher values than those of the IRAsource at angles less than 20° or greater than 160°, whereas both sources show similar trends around 90°.

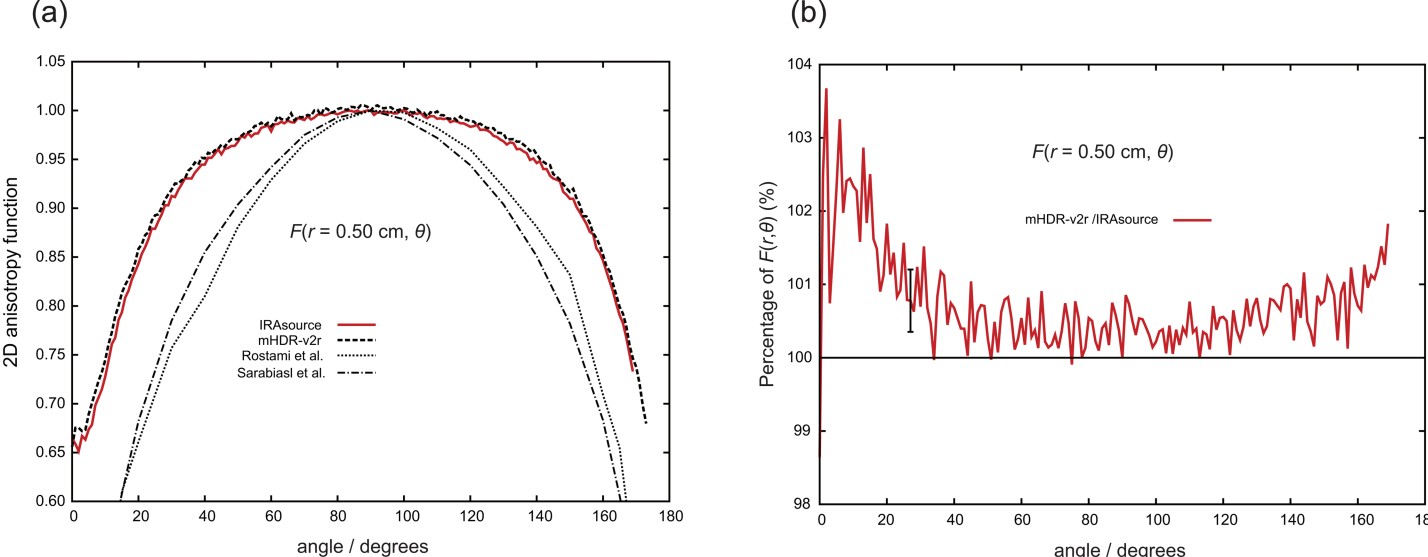

**Fig 9**. **(a) Comparison of two-dimensional (2D) anisotropy functions at distances of 0.50 cm.** (b) The ratio of 2D anisotropy functions to IRAsource at 0.50 cm. The error bar at 30° is derived from the relationship between the IRAsource and the mHDR-v2r [3].

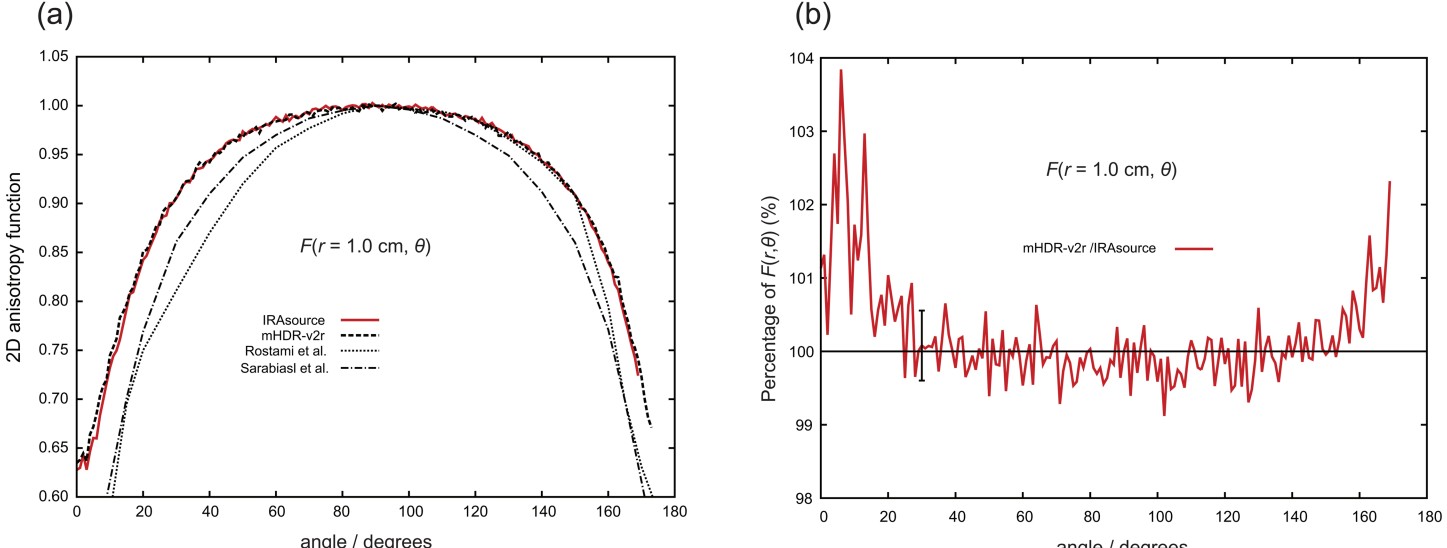

**Fig 10**. **(a) Comparison of two-dimensional (2D) anisotropy functions at distances of 1.0 cm.** (b) The ratio of the 2D anisotropy functions to that of the IRAsource at 1.0 cm. The error bar at 30° was derived from the relationship between the IRAsource and mHDR-v2r [3].

## Discussion

### Applicability of results

With respect to the dose rate constant, the results were consistent within the margin of error. Regarding the radial dose function, the IRAsource results in this analysis showed a shape similar to that of mHDR-v2r [3]. At greater distances (up

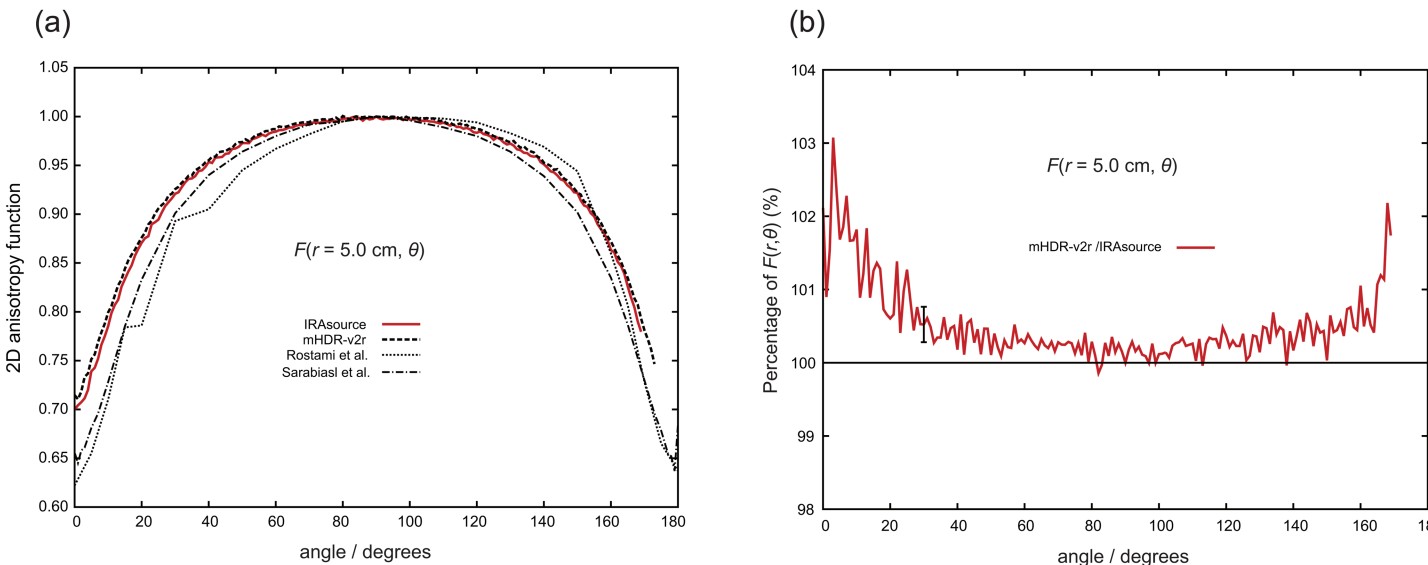

**Fig 11. (a) Comparison of two-dimensional (2D) anisotropy functions at distances of 5.0 cm.** (b) The ratio of 2D anisotropy functions to IRAsource at 5.0 cm. The error bar at 30° was derived from the relationship between the IRAsource and mHDR-v2r [3].

to 10 cm), the influence of the capsule became negligible, and the results for IRAsource and mHDR-v2r converged. Comparison of the 2D anisotropy function with other experimental results for IRAsource showed no agreement. A complete deviation was observed, particularly at distances of less than 1 cm. This deviation is not solely attributed to the omission of the source electron effects. As described in the introduction, the thickness at the tip of the IRAsource capsule was more than twice that of the mHDR-v2r. Accordingly, at $\theta = 0°$, mHDR-v2r is expected to exhibit a higher intensity, whereas at $\theta = 90°$, both sources are anticipated to show nearly equivalent intensities. (To be precise, both the capsule geometry and composition contribute to the difference in results.) The results of this study showed an expected range, from 0.25 cm to 5.0 cm, supporting the reliability of the findings. In this study, an MC simulation was conducted, which is related to the previous mHDR-v2r [3]. The previous results, which are compared in detail with those of Granero et al., demonstrate the validity of the MC simulations. The main program used in this MC simulation is the same one used to calculate dosimetry using mHDR-v2r. During the execution of the main program, the geometry and material composition data from IRAsource were loaded, and calculations were performed. The findings were obtained under robust simulation settings and are therefore considered reliable. Ultimately, accurate absorbed dose intensities were obtained, and these results, especially the parameters of the TG-43U1 protocol, may be helpful in clinical practice. The limitations of this study include the lack of actual IRAsource measurement and calculation results, which prevents comparisons between IRAsources. When comparing the mHDR-v2r dosimetry with that reported by Granero et al., there was a noticeable difference of about 2% [3]. If these results are correct, it is expected that the difference between these IRAsource results and upcoming IRAsource measurements will be within 2%.

## Uncertainties of typical $\dot{D}(0.10\text{cm}, 90°)$, $\dot{D}(1.0\text{cm}, 90°)$, and $\dot{K}(10\text{cm})$

The uncertainty related to the photon spectrum and MC physics was referenced from findings by Rivard et al. [9]. The uncertainty of the $^{192}$Ir electron spectrum based on nuclear decay data for dosimetry calculation (DECDC2) [20], was 0.03%. The source and capsule geometry were referenced from Sarabiasl et al. However, because no reference value for the dose rate at 0.1 cm exists, the uncertainty is assumed to be $\geq 0.75\%$. The uncertainty in volume averaging represents the difference between the median extrapolated from the volume-averaged distance and the value aggregated in

**Table 1. Uncertainty analysis for the IRAsource based on MC simulations.** Types A and B uncertainty components are categorized using stochastic and systematic effects, respectively.

| Component | $\dot{D}(0.10\text{cm}, 90°)$ | | $\dot{D}(1.0\text{cm}, 90°)$ | | $\dot{K}(10\text{cm})$ | |
|---|---|---|---|---|---|---|
| | Type A | Type B | Type A | Type B | Type A | Type B |
| Source:capsule geometry | - | $\geq 0.75\%$ | - | $0.75\%$ | - | $0.75\%$ |
| $^{192}$Ir photon spectrum | - | $1\%$ | - | $1\%$ | - | $1\%$ |
| $^{192}$Ir electron spectrum | - | $0.03\%$ | - | $0.03\%$ | - | $0.03\%$ |
| MC physics | - | $0.05\%$ | - | $0.05\%$ | - | $0.05\%$ |
| Tally volume averaging | - | $0.03\%$ | - | - | - | - |
| Tally statistics | $0.14\%$ | - | $0.19\%$ | - | $0.30\%$ | - |
| Total (k = 1) uncertainty | $\geq 2.00\%$ | | $2.02\%$ | | $2.13\%$ | |

the bin. The uncertainties in the volume averaging at $\dot{D}(1.0 \text{ cm}, 90°)$ and $\dot{K}(10 \text{ cm})$ were $< 0.002\%$. These uncertainties are listed in Table 1. The total uncertainties of $\dot{D}(0.10 \text{ cm}, 90°)$, $\dot{D}(1.0 \text{ cm}, 90°)$, and $\dot{K}(10 \text{ cm})$ were $\geq 2.00\%$, $2.02\%$, and $2.13\%$, respectively. When comparing using MC simulations, the uncertainties in the source and capsule geometry resulting from the manufacturing process can be neglected. In this case, the uncertainties of $\dot{D}(0.10 \text{ cm}, 90°)$ and $\dot{D}(1.0 \text{ cm}, 90°)$ and $\dot{K}(10 \text{ cm})$ were less than $1.4\%$.

## Conclusion

In this study, the dose rate constant, radial dose function, and 2D anisotropy function of the IRAsource were determined using MC simulations, accounting for the source electrons. Comparisons were performed using previously reported data [12–14] and an mHDR-v2r source [3]. The dose rate constant showed good agreement with the reported values within the margin of error, and the radial dose function exhibited trends similar to those of the mHDR-v2r source. Notably, the dose distributions of both sources converged at distances of approximately 10 cm, where the influence of capsule thickness diminishes. The 2D anisotropy function did not perfectly match other experimental results; nonetheless, the findings of this study reflected reasonable trends consistent with differences in source capsule geometry, which are considered reliable within the 0.25 cm to 5.0 cm range. These results provide useful foundational data for dose evaluation in IRAsource. However, to verify these results, further dosimetric results for IRAsources are needed, that is, experimental measurements, and calculations of MC simulations other than MCNP and EGS5 codes. For more accurate calculations, incorporating source electrons with source photons into the MC simulation code is desirable.

## Supporting information

**S1 File. Dose data for IRAsource.** Data supporting the findings of this study are available in the supplementary material of this article. The supplementary material contains detailed dosimetric data, such as source structures, absorbed dose rates, TG-43U1 datasets, and air kerma rates associated with the dose rate constants. The source structure is written in CG so that it can be used with EGS5.
(XLSX)

## Acknowledgments

I would like to thank Editage (www.editage.com) for English language editing.

## Author contributions

**Conceptualization:** Shuhei Tsuji.

**Data curation:** Shuhei Tsuji.

**Formal analysis:** Shuhei Tsuji.

**Investigation:** Shuhei Tsuji.

**Methodology:** Shuhei Tsuji.

**Resources:** Shuhei Tsuji.

**Software:** Shuhei Tsuji.

**Supervision:** Shuhei Tsuji.

**Validation:** Shuhei Tsuji.

**Visualization:** Shuhei Tsuji.

**Writing – original draft:** Shuhei Tsuji.

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
