## [Decision Letter · Decision Letter 0]

21 Aug 2025

PONE-D-25-35311Monte Carlo Dosimetric Characterization of the IRAsource High Dose Rate Iridium-192 Brachytherapy Source: Comparison with the mHDR-v2r ModelPLOS ONE

Dear Dr. Tsuji,

Thank you for submitting your manuscript to PLOS ONE. After careful consideration, we feel that it has merit but does not fully meet PLOS ONE’s publication criteria as it currently stands. Therefore, we invite you to submit a revised version of the manuscript that addresses the points raised during the review process.

1- **Please address the issue of the low energy simulation****2- Adress the issues of using graaphs from previous publication**

We look forward to receiving your revised manuscript.

Kind regards,

Christopher Njeh

Academic Editor

PLOS ONE

Journal Requirements:

Additional Editor Comments:

Please address the issues of the low energy simulations

Reviewers' comments:

Reviewer's Responses to Questions

**Comments to the Author**

1. Is the manuscript technically sound, and do the data support the conclusions?

Reviewer #1: Yes

Reviewer #2: Partly

2. Has the statistical analysis been performed appropriately and rigorously?

Reviewer #1: Yes

Reviewer #2: N/A

3. Have the authors made all data underlying the findings in their manuscript fully available?

Reviewer #1: Yes

Reviewer #2: Yes

4. Is the manuscript presented in an intelligible fashion and written in standard English?

Reviewer #1: Yes

Reviewer #2: Yes

5. Review Comments to the Author

Reviewer #1: 1) The author performed Monte Carlo (MC) simulation close to the IRAsource at sub-millimeter spacial resolution. Please provide detailed description on how MC was implemented in those regions.

2) The results showed that the absorbed dose (from both photon and electron sources) is lower than that from mHDR-v2r. In you Methods section, the author only simulated photons with energy greater than 10 keV. Please justify and explain why 10 keV cut-off energy was used. It is possible that the results of lower absorbed dose in this study is due to the exclusion of low energy photons (<10 keV).

3) Line 119 to 122, the dose rate constant (1.112) from MCNP4C is between that (1.129) from HD-810 film and that (1.084) from EBT film, which contradicts to Line 122, "The value obtained from the MC simulation was higher". Please correct it.

4) Line 167 to 169, how would lack of actual IRA source measurement affects your results. The manuscript's result is purely from Monte Carlo without any experiment validation. The readers need to know how much error would be introduced to the results due to lack of accurate IRA source measurements.

5)Line 199 to 200, In abstract, the author indicates that electron source is incorporated in MC simulation. However, in Line 199 to 200, it shows this incorporation will be future work. Please explain this conflicting information.

Reviewer #2: Major points

• All mHDR-v2r data, methods, and figures are reused from their previous manuscript without citation (https://pubmed.ncbi.nlm.nih.gov/38335156/). Failure to disclose reused material may be considered self-plagiarism.

• Include a side-by-side detailed figure containing both IRAsource and mHDR-v2r. If the source lengths are identical (3.5 mm) and the only structural difference lies in the end-cap thickness (0.53 mm for the IRAsource vs. 0.2 mm for the mHDR-v2r), and given that the IRAsource dosimetric protocol has already been published, the current study does not appear to offer substantial new information or novel insight.

• The manuscript does not include any direct experimental measurement for the IRAsource. While comparisons to published data are performed, the absence of experimental benchmarks for the specific simulation configuration weakens the reliability of the conclusions.

• The study reports higher radial dose function values than other IRAsource MC studies, yet provides only a limited explanation for this discrepancy.

• The anisotropy data shows deviations, particularly in the 0.25–1 cm range. The authors attribute this to capsule differences, but alternative explanations.

Minor points

• Quantifiable results are lacking in the abstract section

• Line 19: The dimensions of 192IR “in IRAsource” are identical to those of mHDR-v2r.

• The conclusion refers to potential clinical utility, but without validation in a treatment planning system or comparison to measured data, this claim is premature.

• Some references lack journal volume/issue/page numbers or are inconsistently formatted.

6. PLOS authors have the option to publish the peer review history of their article (what does this mean?). If published, this will include your full peer review and any attached files.

Reviewer #1: No

Reviewer #2: No

---

## [Author Response · Author response to Decision Letter 1]

22 Sep 2025

I have provided detailed responses to each comment in Response_to_Reviewer.docx.

---

## [Decision Letter · Decision Letter 1]

14 Dec 2025

Monte Carlo Dosimetric Characterization of the IRAsource High Dose Rate Iridium-192 Brachytherapy Source: Comparison with the mHDR-v2r Model

PONE-D-25-35311R1

Dear Dr. Tsuji,

We’re pleased to inform you that your manuscript has been judged scientifically suitable for publication and will be formally accepted for publication once it meets all outstanding technical requirements.

Kind regards,

Mohammadreza Hadizadeh, Ph.D.

Academic Editor

PLOS One

Additional Editor Comments (optional):

Reviewers' comments:

Reviewer's Responses to Questions

**Comments to the Author**

1. If the authors have adequately addressed your comments raised in a previous round of review and you feel that this manuscript is now acceptable for publication, you may indicate that here to bypass the “Comments to the Author” section, enter your conflict of interest statement in the “Confidential to Editor” section, and submit your "Accept" recommendation.

Reviewer #1: All comments have been addressed

Reviewer #3: All comments have been addressed

2. Is the manuscript technically sound, and do the data support the conclusions?

Reviewer #1: Yes

Reviewer #3: Yes

3. Has the statistical analysis been performed appropriately and rigorously?

Reviewer #1: Yes

Reviewer #3: Yes

4. Have the authors made all data underlying the findings in their manuscript fully available?

Reviewer #1: Yes

Reviewer #3: Yes

5. Is the manuscript presented in an intelligible fashion and written in standard English?

Reviewer #1: Yes

Reviewer #3: Yes

6. Review Comments to the Author

Reviewer #1: (No Response)

Reviewer #3: Author addressed all the comments/concerns adequately. Considering this, I believe the paper can now be accepted for publication. I have no comments/concerns.

7. PLOS authors have the option to publish the peer review history of their article (what does this mean?). If published, this will include your full peer review and any attached files.

Reviewer #1: No

Reviewer #3: **Yes:** Mehrdad Shahmohammadi Beni

---

## [Editor Report · Acceptance letter]

PONE-D-25-35311R1

PLOS One

Dear Dr. Tsuji,

I'm pleased to inform you that your manuscript has been deemed suitable for publication in PLOS One. Congratulations! Your manuscript is now being handed over to our production team.

Kind regards,

on behalf of

Dr. Mohammadreza Hadizadeh

Academic Editor

PLOS One